# Assessment of Motion and Muscle Activation Impacts on Low Back Pain during Pregnancy Using an Inertial Measurement Unit

**Saori Morino [1,2,*]**, **Mamoru Yamashita [3]**, **Fumiko Umezaki [3]**, **Hiroko Hatanaka [3]** and **Masaki Takahashi [4]**

[1]  Department of Physical Therapy, Faculty of Comprehensive Rehabilitation, Osaka Prefecture University, Habikino 583-8555, Japan

[2]  School of Science for Open and Environmental Systems, Graduate School of Science and Technology, Keio University, Yokohama 223-8522, Japan

[3]  Kishokai Medical Corporation, Nagoya 460-0003, Japan; ceo@kishokai.or.jp (M.Y.); umezaki@kishokai.or.jp (F.U.); hatanaka@kishokai.or.jp (H.H.)

[4]  Department of System Design Engineering, Faculty of Science and Technology, Keio University, Yokohama 223-8522, Japan; takahashi@sd.keio.ac.jp

*  Correspondence: morino@rehab.osakafu-u.ac.jp; Tel.: +81-72-95-2111

**Abstract:** Specific physiological changes during pregnancy exert excessive strain on muscles such as the erector spinae (ES) and contribute to low back pain (LBP). The link between LBP and sit-to-stand (STS) motion has previously been investigated through motion analysis using an inertial measurement unit (IMU); however, the factors leading to LBP have not been revealed. Moreover, clinicians require an effective assessment method for reducing the physical burden on pregnant women. Therefore, the investigation of the relationships between motion, muscle load calculated from musculoskeletal model for pregnancy, and the severity of LBP during STS in pregnant women was conducted. Furthermore, this study proposes a method for assessing motion and muscle load during STS using an IMU. The relationship among (i) motion evaluation indices and ES muscle torque, and (ii) the ES torque and the intensity of LBP during STS was investigated. As the results, significant positive correlations were observed between (i) the angular velocity of the torso in the sagittal plane and ES torque, and (ii) two types of evaluation indices of ES torque and intensity of LBP. The proposed method by an IMU attached to the torso could effectively assess ES load related to LBP during STS in pregnant women.

**Keywords:** musculoskeletal model; pregnancy; motion analysis; muscle load evaluation; low back pain

## 1. Introduction

Lumbopelvic pain (LPP), which includes low back pain (LBP) and pelvic girdle pain, are common discomforts experienced by women during pregnancy [1]. Symptoms can have a negative impact on the quality of life of the mother in both prenatal and postnatal stages [2,3]. Body weight gain, especially in the abdominal area, and a shift in the position of the body's center of gravity due to fetal growth are distinct changes experienced during pregnancy and a common risk factor of LPP. In addition, an expanded gravid uterus can stretch and weaken the abdominal muscles [4]. These changes in the physical and musculoskeletal system affect the posture and movement of pregnant women and impose strain on muscles and joints, which contributes to pain at various locations such as the lower back [5,6]. For example, the sit-to-stand (STS) motion is a cause of LBP during pregnancy [7]. Hence, physical

methods are required to manage musculoskeletal disturbances during pregnancy, together with the coaching of proper movement to avoid physical loading [8].

Clinical motion analysis typically employs visual observation as the dominant means of assessment; however, more quantitative and objective motion assessments are required to obtain meaningful results [9]. For example, the inertial measurement unit (IMU) allows objective motion measurement without limitations related to the measurement environment or disturbances in motion. Therefore, the IMU has been widely used to analyze motions including gait in a straight path and STS [10,11]. Our previous research investigated the differences in torso motion between pregnant women with and without LPP during STS using an IMU [12]. In the study, an IMU was attached to the lower torso, and angular velocity data in the sagittal plane were used to assess the flexion and extension movements of the trunk. Some key indices were then proposed, that is, maximum peak, minimum peak, and peak-to-peak (PP). The maximum peak represents forward motion of the torso, and the minimum peak represents backward motion, with larger values indicating faster motion. PP is the difference between the maximum and minimum peaks and therefore represents a shift in motion from forward to backward. The results of the study revealed that the maximum peak was greater in the LPP group than in the non-LPP group. However, the role of other motion characteristics as risk factors of pain, such as additional stress on lumbar muscles due to muscle load, remains unclear.

Physical movement therapy based on assessments of motion analysis and muscle load, such as the exerted muscle force, increases the effectiveness of physical therapy for relieving pain. Evaluation of the erector spinae (ES) muscle force is particularly useful for managing LBP because muscle load and fatigue are two of the main risk factors associated with LBP during pregnancy [1]. Although the musculoskeletal model can be used to estimate muscle force, it is difficult to estimate the co-contraction activation of agonist and antagonist muscles, such as the ES and rectus abdominis (RA) [13,14]. To address these issues, we previously proposed a method for estimating muscle torque from the musculoskeletal model using a genetic algorithm (GA) for pregnant women [15].

STS motion assessment should consider the ES muscle load in order to manage LBP in pregnant women. However, the link between muscle load during STS motion and LBP has not yet been revealed for pregnant women. Therefore, the aim of this study was to propose a method for assessing motion and muscle load during STS motion using an IMU. A secondary aim was to investigate the relationship among STS motion, muscle load, and LBP during pregnancy using the proposed assessment method.

## 2. Materials and Methods

### 2.1. Study Design

This is a cross-sectional study. Body motion analysis of STS motion in pregnant women was performed to investigate the relationship among STS motion, muscle load, and LBP. The motion condition was determined using pitch angular velocity data obtained from an IMU attached to the lower trunk of pregnant participants with LBP to evaluate their specific motions [16]. Subsequently, musculoskeletal models tailored to each participant in the study were constructed, and the muscle torque was estimated using a GA. Finally, the different relationships were investigated using the value of motion indices, estimated ES muscle torque, and intensity of LBP during pregnancy. The STROBE checklist of this study is Table S1.

### 2.2. Participants

The study protocol was approved by the Ethics Committee (Kishokai Medical Corporation: approval no. 2015_002). Pregnant women who were undergoing a prenatal health checkup in an obstetrics and gynecology clinic were invited to participate in this study. The purpose and protocol of this study were explained to all participants by the examiner. Written informed consent was obtained from each participant, including consent to participate and to publish the findings. Few studies have focused on the biomechanics of pregnant women and, moreover, studies that investigated the

relationships between biomechanical factor and some complaints by motion analysis for pregnancy could not be found. Then, some studies that focused on the biomechanics of pregnant women in Japan were referenced for determining sample size [17,18]. In each study, eight pregnant women were investigated. Therefore, the sample size of this study was decided as the same number of these previous studies, and thus 11 women were recruited, taking account of the case of dropout. Hence, a group of 11 pregnant women participated in this study. The average duration of pregnancy was 34.5 weeks (standard deviation = 2.0 weeks), and all participants had singleton pregnancies. None of the participants suffered from any serious neurological or orthopedic conditions, particularly those associated with LBP, such as intervertebral disk displacement or spinal cord injury. Before the STS motion analysis, it was confirmed that the participants had not suffered any external injuries that could influence the analysis. Additionally, physical characteristics (age, height, body mass at the time of experiment, body mass gained during pregnancy) were obtained via a questionnaire. Personal data of the participants are listed in Table 1.

**Table 1.** Personal characteristics of the study participants.

| | Total | With LBP During Trial Fast | Without LBP During Trial Fast | *p*-Value |
|---|---|---|---|---|
| | (*n* = 11) | (*n* = 9) | (*n* = 2) | |
| Age (years) | 30.0 (3.1) | 29.4 (2.9) | 32.5 (3.5) | 0.218 |
| Height (cm) | 156.2 (5.8) | 156.6 (6.4) | 154.5 (0.7) | 0.582 |
| Body mass (at the time of experiment) (kg) | 62.1 (10.9) | 61.2 (9.7) | 65.8 (20.2) | 0.727 |
| Body mass (before pregnancy) (kg) | 50.8 (7.3) | 49.7 (6.1) | 55.5 (13.4) | 0.582 |
| Duration of pregnancy (weeks) | 34.5 (2.0) | 34.6 (1.9) | 34.5 (3.5) | 0.999 |
| Number of childbirths before the experiment | 0.7 (0.6) [0–2] | 0.8 (0.7) [0–2] | 0.5 (0.7) [0–1] | 0.727 |
| Number of women with LBP [1] before pregnancy (percentage (%)) | 7 (63.6) | 5 (55.6) | 2 (100.0) | N/A |
| Average intensity of LBP during trial fast | 2.5 (1.9) [0–6] | 3.1 (1.5) [1–6] | 0 | N/A |

[1] LBP: low back pain; Values are mean values (standard deviation) with [range], except for the number of women with LBP before pregnancy. *p*-values are the result of the Mann–Whitney *U* test.

### 2.3. Sensors for the STS Motion Experiment

Motion, force, and electromyographic (EMG) signal data were obtained using the protocol of our previous study to calculate joint torque and estimate muscle torque [15]. An IMU (TSND151, ATR-Promotions Co., Ltd., Kyoto, Japan) was used to obtain joint angles and segment orientations during the STS motion analysis. The IMU contains tri-axial gyroscopes, accelerometers, and magnetometers and has been used for human motion analysis [19,20]. In addition, the IMU can be used in synchronization with the EMG electrodes that were used in this study. The participants were assessed using six IMUs. Five IMUs were bilaterally attached to a fixed belt, 10 cm above the lateral malleolus, 10 cm above the patella, and at the level of the L4 spinous process in order to measure the bilateral motion of the shank segments, thigh segment, and lower trunk (pelvis), respectively (Figure 1a). Another IMU was attached to the level of the C7 spinous process using skin tape to measure the motion of the upper trunk. A Nintendo Wii balance board (WBB; Nintendo, Kyoto, Japan) was used to measure the vertical reaction forces (Figure 1b). WBB has been recently proposed as an easily available device for measuring ground reaction force and center of pressure (CoP) displacement and an inexpensive alternative to force plates, which are common in laboratories for the assessment of posture and balance [21,22]. To measure the reaction force from the chair and the ground reaction force, respectively, one WBB was placed on the chair, and the other was placed under the participant's feet. Four surface EMG electrodes (SE-C-AMP-H100, ATR-Promotions Co., Ltd., Kyoto, Japan) were used to measure muscle activation of the ES and RA muscles (Figure 1a). Electromyography electrodes

were bilaterally placed in parallel with the fiber orientation of the RA and longissimus (part of the ES) muscles.

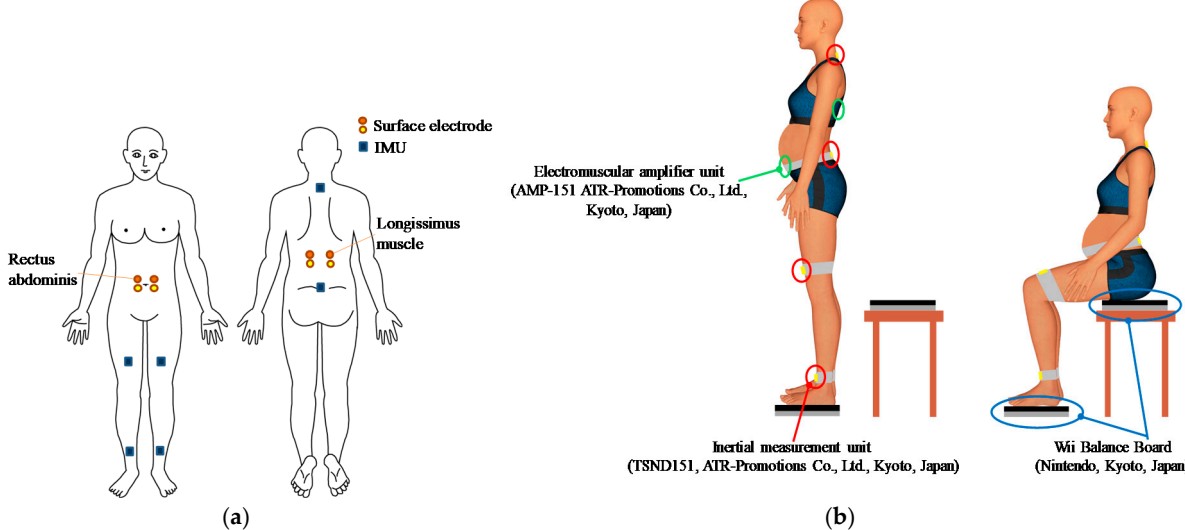

**Figure 1.** Experimental settings: (**a**) position of the inertial measurement units (IMUs) and surface electrodes; (**b**) experimental settings.

### 2.4. Participant-Specific Musculoskeletal Models

Full-body musculoskeletal models were constructed for each pregnant woman using Biomechanics of Bodies (BoB; Marlbrook Ltd., Bromsgrove, United Kingdom). BoB is a package containing the musculoskeletal model of a human being, 36 skeletal segments, and over 600 muscle units [23]. BoB's biomechanical software is based on the MATLAB and Simulink environments [24]. Torque in the joints corresponding to the observed motion and force data was calculated by inverse dynamic analysis using BoB, and the muscle torque was estimated using an optimization approach [25,26]. The following process was then undertaken to develop musculoskeletal models for each subject in accordance with our previously proposed procedure [15].

A skeletal model previously modified to the average body type of Japanese women was used [15]. Then, the total body height and weight were adjusted for each participant. Furthermore, the muscular model was also changed because the abdominal muscles were stretched and weakened during pregnancy by an enlarged gravid uterus [15]. In the previous study, a model of the stretched RA muscle was constructed using information from an anthropometric dimensions database for pregnant women and measurements of the three height levels of waist girth for the participants. Thus, the same three height levels of waist girth were measured for each participant in this study. Weight gain, which is a characteristic change during pregnancy, was also considered for each participant.

### 2.5. Experimental Settings and STS Motion Conditions

An overview of the experimental settings is given in Figure 1b. For the motion task, participants with IMUs attached to their body segments were asked to start from a standing position in front of a chair. Then, they were instructed to sit down and stand up using the chair as a support. On the basis of the results of our previous study [12], two types of STS motion, characterized by different values of maximum peak angular velocities of the torso in the sagittal plane (slow and fast), were conducted three times by each participant. The difference between the two types of STS motion was the forward incline speed of participant trunks. The incline speed was checked and instructed by the examiner. The participants moved at the prescribed velocities and performed the STS motion without using hands. The participants were asked to put their arms along their sides. The maximum peak represents the maximum velocity of forward trunk movement during STS motion (an example of the indices from

our previous research is shown in Figure A1). Specifically, the participants were instructed to move at low angular velocities for "trial slow" and at high angular velocities for "trial fast", so that the IMU attached to the lower trunk would indicate a value similar to that given in Table A1. The order of the type of motion was determined randomly. A rest period of approximately 10 s was fixed between each motion. An armless, backless chair was used and adjusted to the height of the participant's knee in the condition with WBB, in accordance with previous research [27]. The STS movement was performed by the participants without their shoes on, and the medial borders of their feet were placed 10–15 cm apart.

### 2.6. Assessment of LBP

During the first visit, the occurrence of LBP in the year prior to the pregnancy was investigated. Then, the participants were asked about the presence or absence of LBP during the STS motion analysis in every trial. If the participants reported LBP, the pain intensity was assessed using an 11-point numerical rating scale (NRS) [28], where the low and high endpoints represent the extremes of "no pain" and "worst pain", respectively. On the basis of the ratings, NRS > 0 was defined as the condition for the presence of LBP. The NRS score during STS motion was obtained for each participant.

### 2.7. Processing of Measured Data

Signal processing of IMU data during STS motion analysis was performed using MATLAB Release 2016a (MathWorks, Release 2016a, Natick, MA, USA). The coordinate system in the experimental settings is shown in Figure 2. First, the segment orientation and joint angles of each segment were calculated as motion data using IMU data and the same protocol as our previous study [15]. The calculated orientations of body segments and joint angles are shown in Figure 2. The direction of the axis of each IMU was determined as the line of each segment.

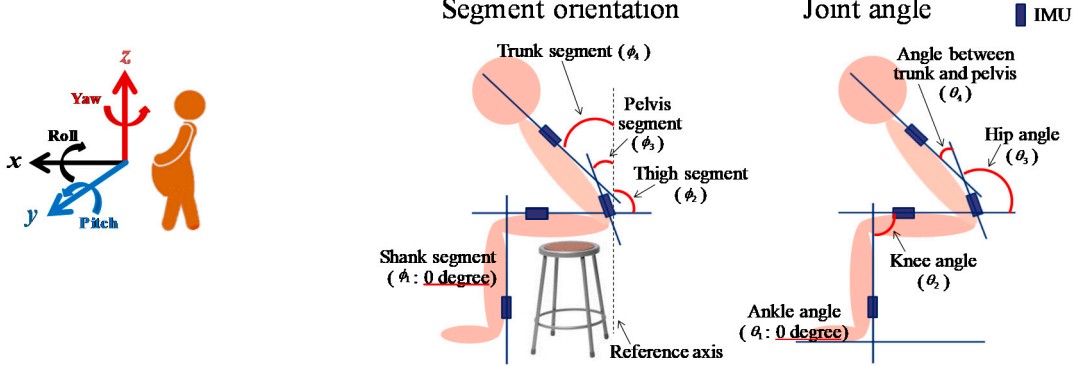

**Figure 2.** Coordinate system (**left**) and definition of orientations and angles (**right**). Calculated orientations of body segments and joint angles from each IMU are shown, where $\theta$ and $\phi$ correspond to the symbols in Equation (1).

The joint angle was defined as 0° for an anatomically upright posture. The vertical axis in the sagittal plane was used as the reference axis; thus, it was defined as 0° to calculate the inclination angle. Joint angles were calculated using Equation (1), where $\theta$ and $\phi$ correspond to the symbols used in Figure 2.

$$
\begin{aligned}
\theta_1 &= \varphi_1 \\
\theta_2 &= 180 + \varphi_2 - \varphi_1 \\
\theta_3 &= \varphi_3 - \varphi_2 \\
\theta_4 &= \varphi_4 - \varphi_3
\end{aligned}
\tag{1}
$$

Next, the ground reaction forces were calculated using data from the WBBs (body mass acting on the WBBs, kg) under the foot and hip. Thus, the force (N) in each trial was calculated using Equation (2). Then, these motion and force data were inputted to the BoB package with the modified musculoskeletal

models for pregnant women in order to calculate the joint torque. Finally, the optimization process in BoB was performed using the joint torque to estimate the muscle torque in the model.

$$F = ma \qquad (2)$$

Here, $F$ is the reaction force (N), $m$ is the body mass measured by the WBB (kg), and $a$ is the acceleration due to gravity (m/s$^2$).

### 2.8. Co-Contraction Activation Estimation Using GA

Muscle torque as an index of muscle load was calculated from obtained data in this study. An overview of the system used to estimate muscle torque is presented in Figure 3. For each trial, muscle torque was estimated by considering co-contraction activation using a GA according to the following process. The procedure for estimating RA and ES muscle torque is fundamentally the same as that in our previous study [15]. First, Equation (3) was used to estimate the muscle torque. The equation is a mathematical model proposed by Oyong and Jauw [29] that converts EMG signals to muscle torque:

$$\tau_{est}(k) = x_1 E(k)^{x_2} + x_3 E(k)^{x_4} \qquad (3)$$

where $k$ is the sampling time, $\tau_{est}(k)$ is the estimated torque, $E(k)$ is the processed EMG signal, and $x_j$ represents the associated model parameters. Accordingly, the torque of each RA and ES muscle, as well as the sum of the two muscle torque results, were calculated using Equation (4):

$$\tau_{est\_RA}(k) = x_1 E_{RA}(k)^{x_2} + x_3 E_{RA}(k)^{x_4}$$
$$\tau_{est\_ES}(k) = x_5 E_{ES}(k)^{x_6} + x_7 E_{ES}(k)^{x_8} \qquad (4)$$
$$\tau_{est}(k) = \tau_{est\_RA}(k) + \tau_{est\_ES}(k)$$

where $E_{RA}$ is the processed EMG signal of RA, and $E_{ES}$ is the processed EMG signal of ES.

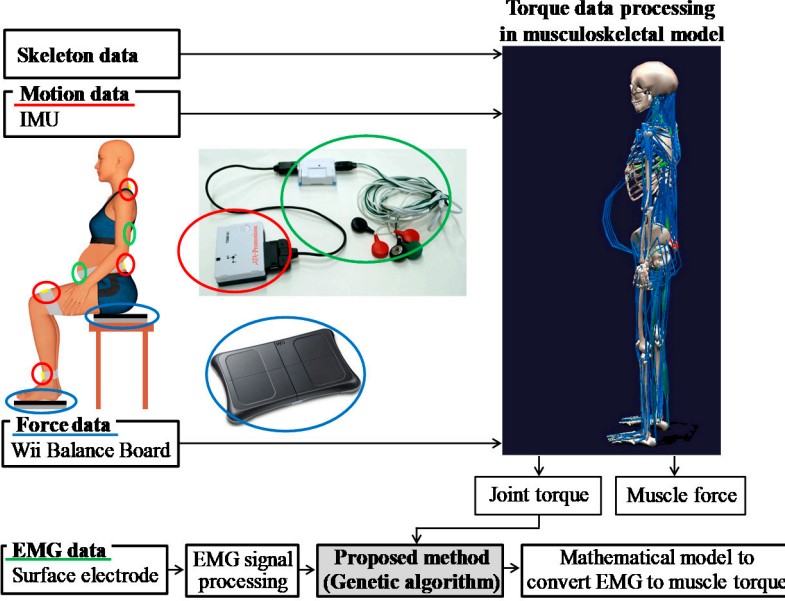

**Figure 3.** Overview of the system used to estimate muscle torque considering co-contraction activation using a genetic algorithm (GA). EMG: electromyography, IMU: inertial measurement unit.

Trial conditions in the GA process such as population size were the same as those in our previous study [15]. Subsequently, the joint torque during STS motion was calculated using the motion, force, and skeletal data via inverse dynamic calculation in the pregnant musculoskeletal model, and the

value of torque was used as the actual torque in the GA. These values enabled us to estimate the muscle torque via an optimized GA using Equation (5). The fitness values ($V_{FS}$) in this equation can be maximized only when a certain estimated muscle torque ($\tau_{est}$) fits the actual joint torque ($\tau_{act}$) as a supervisory signal.

$$V_{FS} = \frac{1}{\sum_{k=1}^{n} \left( \tau_{act}(k) - \tau_{est}(k) \right)^2} \tag{5}$$

Here, $V_{FS}$ is the fitness value, $n$ is the number of $\tau_{act}$ and $\tau_{est}$ values in each experiment trial, and $\tau_{act}$ is the joint torque calculated from the musculoskeletal model. In addition, a completion condition was set in this study. To assess the precision of the estimated muscle torque, the root mean square error (RMSE) was calculated using Equation (6):

$$d_{RMSE} = \sqrt{\frac{1}{n} \sum_{k=1}^{n} \left( \tau_{act}(k) - \tau_{est}(k) \right)^2} \tag{6}$$

where $\tau_{act}$ is the actual joint torque during movement, and $\tau_{est}$ is the estimated muscle torque. Smaller RMSE values indicate that the muscle torque values were better estimated. From a previous study that estimated lumbar muscle force using the EMG-based model [30], the mean ratio of RMSE between the measured and estimated lumbar sagittal moments to the maximum peak of the measured sagittal moments was approximately 22.34% in the sagittal plane. In the study, RMSE values that were higher than those obtained by previous research were not considered suitable for estimated muscle torque [31,32]. Thus, the cut-off completion condition value in this study was 20% of the peak value of $\tau_{act}$ in each trial. The GA step was completed when the value of RMSE was less than 20% of the $\tau_{act}$ value in each trial.

*2.9. Evaluation Index of Muscle Torque*

A universal evaluation index calculated from muscle torque during STS motion has not been established by previous research. Thus, the following indices were calculated to investigate the relationship between the motion features and muscle torque of the ES during STS motion in pregnant women (Figure 4). First, the value of the ES muscle torque (MT) during STS motion at the maximum peak of the angular velocity in the sagittal plane of the lower trunk segment (MP) was detected and termed MT-MP (Figure 4, point A). Then, the difference in muscle torque at MP and the standing position (DMT-MP-stand) was also calculated (Figure 4, point B) to remove the individual differences in muscle torque in the standing position. The maximum peak of the ES muscle torque during STS motion (Max-MT) was also detected (Figure 4, point C), and the difference between the value of ES muscle torque during STS motion and the standing position (DMT-Max-stand) was calculated (Figure 4, point D). Furthermore, the root mean square (RMS-MT), which is commonly used to express the effective value of a waveform, was calculated using the ES muscle torque data for each motion by Equation (7). All relevant indices were calculated for each trial and each participant.

$$d_{ES\_RMS} = \sqrt{\frac{1}{n} \sum_{k=1}^{n} \left( \tau_{est\_ES}(k) \right)^2} \tag{7}$$

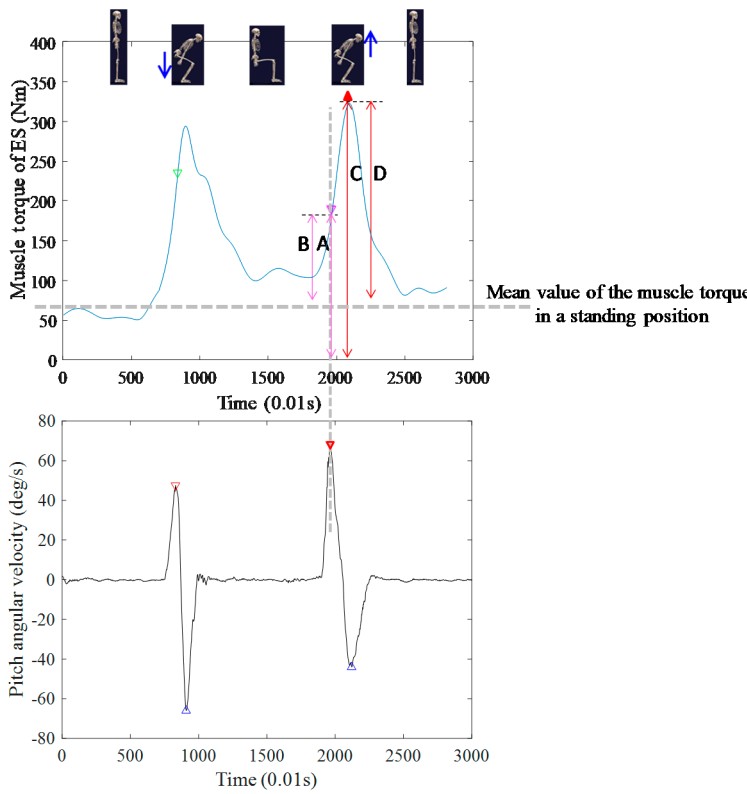

**Figure 4.** Indices for muscle torque evaluation.

*2.10. Statistical Analysis to Compare Muscle Torque Evaluation Indices*

The relationship among the indices of angular velocity, which represents motion information, the estimated ES muscle torque, and LBP, was investigated using participant data from the motion analysis. Before the investigation, a normality of each of the data was checked by using of a Shapiro–Wilk test. Then, a non-parametric test was conducted because none of the datasets followed a normal distribution. First, a Wilcoxon signed-rank test was conducted to investigate the differences among the five indices of the estimated ES muscle torque (MT-MP, DMT-MP-stand, Max-MT, DMT-Max-stand, and RMS-MT) between the STS motion in "trial slow" and "trial fast". That is, the five indices were compared for all 11 participants in the two trials.

Next, a bivariate correlation analysis using the Spearman's rank correlation coefficient was conducted to investigate the correlation between the five indices of estimated ES muscle torque and the maximum peak during STS motion. In this analysis, PP was investigated instead of the maximum peak because the timing of Max-MT was almost the same as PP among the motion indices, as observed in Figures 4 and A1. Moreover, the data from "Trial fast" representing the characteristic motion of pregnant women who were experiencing pain were used to investigate the motions that could be linked to LBP. Furthermore, Spearman's rank correlation coefficients between each index of the estimated ES muscle torque and the maximum peak during STS motion, as well as between each index and PP value during STS motion, were calculated. Finally, a Spearman's rank correlation coefficient was calculated using five indices of the estimated ES muscle torque and the LBP-related NRS score to investigate the correlation between the degree of muscle load and the intensity of LBP. Nine pregnant women reported LBP during "trial fast," but none reported LBP during "trial slow". Thus, data of these nine participants were used in the last statistical analysis. The analyses were performed on SPSS 23.0 (IBM SPSS Statistics, Chicago, IL, USA) with a significance threshold of $p > 0.05$.

## 3. Results

### 3.1. Difference of Muscle Torque According to Difference of Motion Trial

The results of the Wilcoxon signed-rank test are presented in Table 2. The DMT-Max-stand index of "trial fast" was significantly higher than that of "trial slow." No significant differences were observed for the other muscle torque indexes between each trial.

**Table 2.** Differences in evaluated indices of muscle torque according to type of sit-to-stand (STS) motion (*N* = 11).

| Evaluation Indices of Muscle Torque | Trial Fast (Nm) | Trial Slow (Nm) | *p*-Value |
|---|---|---|---|
| MT-MP [1] | 187.0 (43.2) | 163.0 (53.6) | 0.213 |
| DMT-MP-stand [2] | 128.5 (44.5) | 96.3 (36.1) | 0.110 |
| Max-MT [3] | 257.2 (54.3) | 212.4 (82.2) | 0.062 |
| **DMT-Max-stand [4]** | **198.2 (60.1)** | **143.9 (78.7)** | **0.041** |
| RMS-MT [5] | 201.7 (34.7) | 189.3 (72.7) | 0.477 |

[1] MT-MP: the value of the erector spinae (ES) muscle torque (MT) during STS motion at the maximum peak (MP) of the angular velocity; [2] DMT-MP-stand: the difference in muscle torque at MP and the standing position; [3] Max-MT: the maximum peak of the ES muscle torque during STS motion; [4] DMT-Max-stand: the difference between the value of ES muscle torque during STS motion and the standing position; [5] RMS-MT: the root mean square of the ES muscle torque. Values shown are mean values (standard deviation). The data in bold is statistically significant.

### 3.2. Correlation between Indices of Estimated Muscle Torque and the Value of Motion Indices in STS

Table 3 lists the results of the statistical correlation analysis performed for the five indices of the estimated ES muscle torque and values of the motion index, revealing that a significant positive correlation was observed between DMT-Max-stand and PP. No significant correlations were observed between maximum peak and indices of estimated muscle torque.

**Table 3.** Correlation analysis results for motion and muscle torque (*N* = 11).

| Evaluation Indices of Muscle Torque | | *r* [1] | *p*-Value |
|---|---|---|---|
| Maximum peak | Max-MT [2] | 0.073 | 0.688 |
| | DMT-Max-stand [3] | 0.120 | 0.507 |
| | MT-MP [4] | 0.011 | 0.950 |
| | DMT-MP-stand [5] | 0.121 | 0.517 |
| | RMS-MT [6] | 0.112 | 0.536 |
| PP [7] | Max-MT | 0.282 | 0.112 |
| | **DMT-Max-stand** | **0.355** | **0.043** |
| | MT-MP | 0.141 | 0.434 |
| | DMT-MP-stand | 0.145 | 0.419 |
| | RMS-MT | 0.176 | 0.327 |

[1] *r*: correlation coefficient; [2] Max-MT: the maximum peak of the ES muscle torque during STS motion; [3] DMT-Max-stand: the difference between the value of ES muscle torque during STS motion and the standing position; [4] MT-MP: the value of the ES muscle torque (MT) during STS motion at the maximum peak (MP) of the angular velocity; [5] DMT-MP-stand: the difference in muscle torque at MP and the standing position; [6] RMS-MT: the root mean square of the ES muscle torque; [7] PP: peak-to-peak. The data in bold is statistically significant.

### 3.3. Correlation between the Indices of Estimated Muscle Torque and the Intensity of LBP in STS

Table 4 presents the results of the statistical correlation analysis for the five indices of estimated ES muscle torque and the LBP-related NRS score, revealing significant positive correlations between (i) Max-MT and LBP intensity and (ii) DMT-MP-stand and LBP intensity. No other significant correlations were observed between the indices of estimated muscle torque and LBP intensity.

**Table 4.** Correlation analysis results for muscle torque and low back pain (*n* = 9).

| Evaluation Indices of Muscle Torque | *r* [1] | *p*-Value |
|:---:|:---:|:---:|
| **Max-MT [2]** | **0.743** | **0.022** |
| **DMT-Max-stand [3]** | **0.726** | **0.027** |
| MT-MP [4] | 0.143 | 0.382 |
| DMT-MP-stand [5] | 0.367 | 0.331 |
| RMS-MT [6] | 0.376 | 0.319 |

[1] *r*: correlation coefficient; [2] Max-MT: the maximum peak of the ES muscle torque during STS motion; [3] DMT-Max-stand: the difference between the value of ES muscle torque during STS motion and the standing position; [4] MT-MP: the value of the ES muscle torque (MT) during STS motion at the maximum peak (MP) of the angular velocity; [5] DMT-MP-stand: the difference in muscle torque at MP and the standing position; [6] RMS-MT: the root mean square of the ES muscle torque. The data in bold is statistically significant.

## 4. Discussion

This study proposed a method for assessing motion and muscle load in pregnant women during STS motion on the basis of an inertial measurement unit. This study also investigated the relationships between motion evaluation indices, muscle load (erector spinae activation), and the severity of LBP. The link between muscle load during STS motion and LBP in pregnant women has not previously been determined, despite this being a common discomfort experienced by women during pregnancy, which can have a negative impact on quality of life during both prenatal and postnatal stages. As its results, this study established the relationship between erector spinae muscle torque and LBP severity during STS motion in pregnant women.

### 4.1. Difference of Muscle Torque According to Difference of Motion Trial

The results in Table 2 suggest that the ES muscle torque during STS motion, excluding muscle torque in a standing still position, was higher for specific motions exhibited by pregnant participants with LBP than for those exhibited by participants without LBP. These results indicate that higher muscle load from an immobile standing position could be related to LBP during STS motion.

### 4.2. Correlation between Indices of Estimated Muscle Torque and the Value of Motion Indices in STS

The positive correlation between DMT-Max-stand and PP, which is a parameter describing the change in incline speed from forward (maximum peak) to backward (minimum peak) motion, suggests that ES muscle load might increase with a higher degree of shift change from forward bending to backward bending motion. Therefore, the results confirm that the value of maximum ES muscle torque in relation to the motion of standing up can be evaluated using the PP of angular velocity from the IMU attached to the lower torso, without the need for a direct evaluation of muscle load.

### 4.3. Correlation between the Indices of Estimated Muscle Torque and the Intensity of LBP in STS

In addition to Table 4, scatter diagrams of the NRS score with the Max-MT and DMT-MP-stand are shown in Figure A2. The significant positive correlation between Max-MT and LBP intensity implies that higher values of maximum ES muscle torque during STS motion could indicate a higher correlation with LBP. Moreover, a significant positive correlation between DMT-MP-stand and LBP intensity was also observed. Thus, a larger value of maximum ES muscle torque in relation to the standing up motion might be related to more severe LBP symptoms.

### 4.4. Relationships among the Motion Indices from an IMU, Estimated Muscle Torque, and the Intensity of LBP in STS

Our previous research revealed that the STS motion particular to pregnant women with LBP can be observed using an IMU attached to the lower torso. For example, a higher maximum angular velocity peak in the sagittal plane was observed in the LBP group. However, other risk factors, such as muscle fatigue, were not considered in that research. In the current study, however, the relationships

among the motion index, muscle torque, and LBP during STS motion were investigated for pregnant women using motion, force, and EMG data, respectively. The results in Section 3.2 reveal a statistically significant positive correlation between PP and ES muscle torque (out of muscle torque in the standing position) during STS motion. Simultaneously, the results in Section 3.3 reveal a statistically significant positive correlation between the maximum muscle torque and the intensity of LBP. These results suggest that the PP of angular velocity in the sagittal plane, obtained from an IMU attached to the lower trunk, may have expressed not only the trunk motion characteristics, but also the ES muscle torque exhibited during STS motion for pregnant women. Mechanical information, such as muscle torque estimated by the proposed model, is more useful for quantifying the physical burden related to LBP. In addition, the results reveal the relationship between higher ES muscle torque and LBP severity during STS motion in pregnant women. Thus, this study indicates the potential for conducting pregnancy motion analysis and muscle torque evaluation, which may be associated with LBP, using the proposed IMU-based method, without the need for EMG measurement.

This is the first study that has focused on the biomechanics of pregnant women and investigated the relationships between ES muscle torque during motion and LBP by motion analysis. However, there are various factors that should be considered when investigating the relationships between biomechanics and LBP during pregnancy. For example, the iliopsoas muscle has a relevant role in STS movement and is related to LBP. Moreover, this muscle might be stretched as well as the rectus abdominis during pregnancy. Similarly, other muscles such as the diaphragm and other abdominal wall muscles might be related to LBP [33,34]. Moreover, other studies have indicated that there is a relationship between myofascial trigger points in the paraspinal muscles and LBP [35]. Therefore, further study would be required to investigate the existence of an extensive and detailed biomechanical factor related to LBP during pregnancy.

## 5. Conclusions

Experiments on STS motion with pregnant women indicated the possibility that muscle torque related to LBP can be evaluated using an IMU. Higher ES muscle load from an immobile standing base state might be related to LBP during STS motion in pregnancy. However, measuring muscle load is difficult, especially during STS motion and for pregnant women. Our results revealed that the PP of angular velocity in the sagittal plane measured by an IMU attached to the lower trunk indicates greater ES muscle torque during STS motion. In addition, our experiments revealed the relationship between higher ES muscle torque and LBP severity during STS motion in pregnancy.

The limitations of this study include the small sample size and cross-sectional design. Therefore, the causal relationship between motion characteristics, exerted ES muscle torque, and LBP during STS motion could not be revealed. Moreover, the value of correlation coefficient of the relationship between PP and DMT-Max-stand was not high. Thus, the correlation might not be strong. Similarly, the results of the relationships between muscle torque and intensity of LBP were not enough to decide the higher muscle torque indicating the higher intensity of LBP. Therefore, other factors should be considered for utilizing the results of this study. Hence, further research that includes a larger number of participants, a longitudinal design, and surveying of other risk factors of LBP is required to support the results of this study. Despite these limitations, this study established the effectiveness of an IMU attached to the torso for assessing STS motion in pregnant women. Moreover, the proposed physical assessment method can be employed to manage LBP during pregnancy.

**Supplementary Materials:** The following are available online at http://www.mdpi.com/2076-3417/10/11/3690/s1, Table S1: STROBE Checklist.

**Author Contributions:** Conceptualization, S.M. and M.T.; methodology, S.M., F.U., and M.Y.; software, S.M.; formal analysis, S.M. and M.T.; investigation, S.M. and H.H.; resources, M.Y., F.U., and M.Y.; data curation, S.M., F.U., and H.H.; writing—original draft preparation, S.M.; writing—review and editing, M.T.; visualization, S.M.; supervision, M.Y.; project administration, M.Y. and M.T.; funding acquisition, S.M. All authors have read and agreed to the published version of the manuscript.

**Funding:** This study was supported by JSPS KAKENHI under grant 20K19348.

**Acknowledgments:** We are grateful to the staff of Kishokai for recruiting the participants and their cooperation in obtaining the assessments. We are also grateful to the pregnant women for their cooperation and participation.

**Conflicts of Interest:** The authors declare no conflict of interest.

## Appendix A

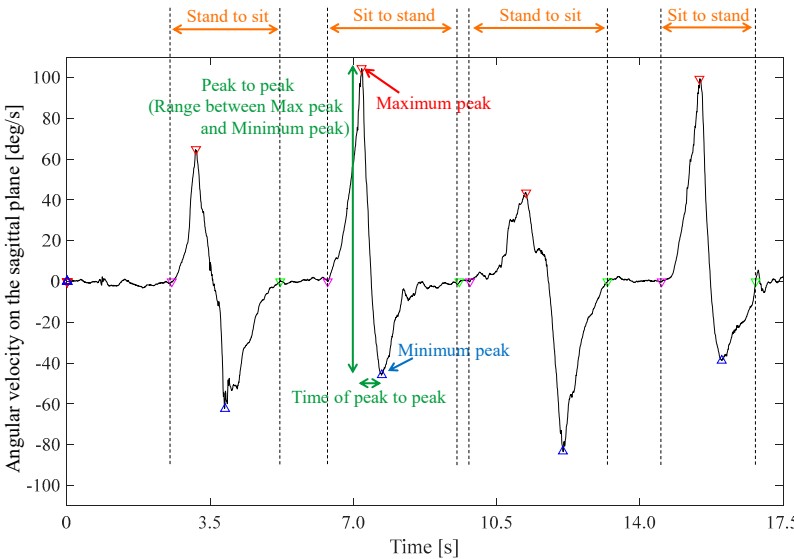

**Figure A1.** Wave forms and indices of angular velocity of lower trunk on the sagittal plane.

**Table A1.** Value of angular velocity of trunk motion in the sagittal plane, according to the occurrence of lumbopelvic pain (LPP) in our previous research.

|  | LPP During Stand Up (N = 10) | Non LPP During Stand Up (N = 19) | *p*-Value |
|---|---|---|---|
| Maximum peak (deg/s) | **88.49 (13.35)** | **73.98 (15.72)** | **0.020** |
| Minimum peak (deg/s) | −43.53 (12.83) | −36.77 (10.79) | 0.144 |
| PP (deg/s) | **132.02 (15.10)** | **110.75 (22.37)** | **0.012** |

A Student's *t*-test was used; values are shown as mean (standard deviation); LPP: lumbopelvic pain, PP: peak-to-peak. The data in bold is statistically significant.

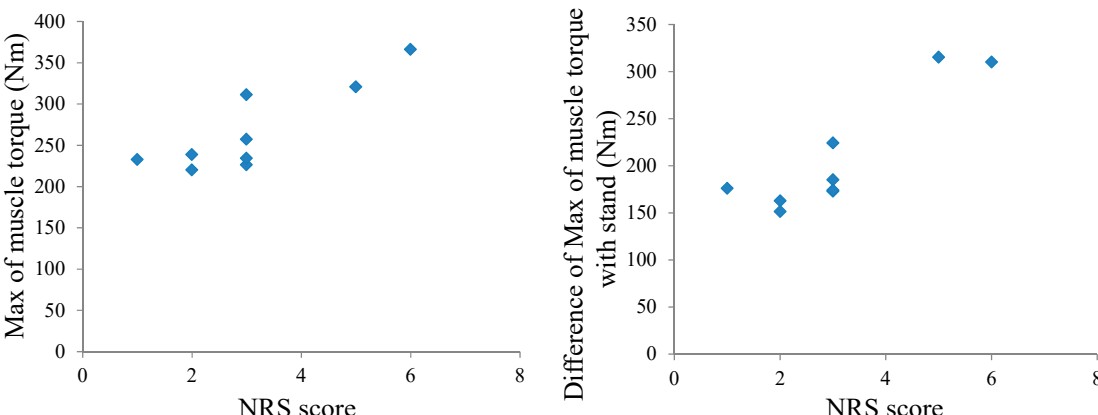

**Figure A2.** Scatter diagram of numerical rating scale (NRS) score with maximum of muscle torque and difference of maximum of muscle torque with stand.

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
