# Peer review of "Assessment of Motion and Muscle Activation Impacts on Low Back Pain during Pregnancy Using an Inertial Measurement Unit"

_applsci, doi:10.3390/app10113690_

Round 1

Reviewer 1 Report

Thanks for yhe opportunity to review this manuscript. Please, consider the following recommendations:

-Abstract: According to the Journal`s recommendations, the abstract should be divided into background, methods, results and discussion. 

-Methods: A study design section should be added detailing the sutdy type as well as the followed recommended guideliness. In addition, provide the recommended checklist of STROBE according to a case-control study available at: https://www.equator-network.org/

-Methods: Table 1: Please expand LBP at footnotes of the table.

-Methods: Please add reliability and validity of the IMU (TSND151, 101 ATR-Promotions Co., Ltd., Kyoto, Japan). The minimum detectable change and its discussion would be useful.

-Methods: The statistical analysis should include the used test for normality analyses. 

-Tables: please, add footnotes with expanded abbreviations from table 2 to table 4.

-Methods: please, add a sample size calculation in order to justify your sample.

-Discussion: Please, discuss about the possible presence of myofascial trigger points in the paraspinal muscles due to their possible influence on paraspinal muscle characteristics (Medicine (Baltimore). 2017 Mar;96(10):e6287. doi: 10.1097/MD.0000000000006287). Please, also discuss about the possible influence of other muscles such as diaphragm (Phys Ther Sport. 2019 May;37:128-137. doi: 10.1016/j.ptsp.2019.03.015.) and abdominal wall (J Orthop Sports Phys Ther. 2013 Jan;43(1):11-9. doi: 10.2519/jospt.2013.4450.) in low back pain during pregnancy.

Author Response

Thank you for your comments regarding our manuscript. We received your comments on May 11, 2020 and we have revised the manuscript in accordance with your comments. Please check the attachment. We hope the changes meet with your approval.

Reviewer 2 Report

Dear Authors, 

the manuscript you sumbitted constitutes a good piece of work

A few concerns follow:

1) Many brands and models of sensors and accelerators are available on the market, but a few of these achieve scientifica validation and results are presented in peer-review papers. In your manuscript you did not mention any previous validation study. Furthermore, you use the Nintendo Wii board for the assessment of reaction force. Please provide some scientific evidence concerning the use of these sensors.

2) In your analysis you did not mention the ileopsoas muscle, which has a relevant role in STS movement and especially during pregnancy this muscle may be stretched, as well as the rectus abdminis (as you reported). Please, add a comment on this.

3) It would be interesting to understand whether a correlation exists between the duration of pregnancy and movement indices.

4) From your description of statistical analysis is not clear wether you performed univariate or multivariate correlations. This would be useful to fully understand the results

Author Response

(The authors gave the same response as above.)

Reviewer 3 Report

Major revisions:

The authors need to address the week correlations, especially pertaining to PP and DMT-max-stand (r <0.4). Although the correlation value is not likely due to randomness (p<0.05), the correlation is not strong enough to indicate to this reviewer that IMU PP can be used to estimate muscle torque. R>0.7 indicates a better correlation between LBP and muscle torque, but this is only 55% explained variance. So the authors need to address what might explain the remaining 45% variance.

The first sentence of the conclusions is most problematic since this study never conducted as analysis between IMU and LBP.

Minor revisions:

  1. Line 123: put a space in "biomechanicalsoftware"
  2. Line 145: change "velocity" to "velocities"
  3. Table 4: seems like you are bolding only statistically significant values. Why is the last row bold?

Author Response

(The authors gave the same response as above.)

Round 2

Reviewer 1 Report

Thanks for addressing all my comments.

Reviewer 3 Report

no additional comments